# Predictive Values of Procalcitonin and Presepsin for Acute Kidney Injury and 30-Day Hospital Mortality in Patients with COVID-19

**DOI:** 10.3390/medicina58060727

**Published:** 2022-05-28

**Authors:** Sin-Young Kim, Dae-Young Hong, Jong-Won Kim, Sang-O Park, Kyeong-Ryong Lee, Kwang-Je Baek

**Affiliations:** 1Department of Emergency Medicine, Konkuk University Medical Center, Seoul 05030, Korea; 20110096@kuh.ac.kr; 2Department of Emergency Medicine, Konkuk University School of Medicine, Seoul 05030, Korea; 20130296@kuh.ac.kr (J.-W.K.); empso@kuh.ac.kr (S.-O.P.); lkrer@kuh.ac.kr (K.-R.L.); kjbaekmd@kuh.ac.kr (K.-J.B.)

**Keywords:** COVID-19, acute kidney injury, mortality, procalcitonin, presepsin

## Abstract

*Background and Objectives*: Acute kidney injury (AKI) is a common complication in patients with coronavirus disease 2019 (COVID-19). We investigated the values of procalcitonin (PCT) and presepsin (PSS) for predicting AKI and 30-day hospital mortality in patients with COVID-19. *Materials and Methods*: We retrospectively evaluated 151 patients with COVID-19 who were admitted to the hospital via the emergency department. The diagnosis of AKI was based on the Kidney Disease: Improving Global Outcomes clinical practice guidelines. *Results*: The median patient age was 77 years, and 86 patients (57%) were male. Fifty-six patients (37.1%) developed AKI, and 19 patients (12.6%) died within 30 days of hospital admission. PCT and PSS levels were significantly higher in patients with AKI and non-survivors. The cutoff values of PCT levels for predicting AKI and mortality were 2.26 ng/mL (sensitivity, 64.3%; specificity, 89.5%) and 2.67 ng/mL (sensitivity, 68.4%; specificity, 77.3%), respectively. The cutoff values of PSS levels for predicting AKI and mortality were 572 pg/mL (sensitivity, 66.0%; specificity, 69.1%) and 865 pg/mL (sensitivity, 84.6%; specificity, 76.0%), respectively. *Conclusion*: PCT and PSS are valuable biomarkers for predicting AKI and 30-day hospital mortality in patients with COVID-19.

## 1. Introduction

Coronavirus disease 2019 (COVID-19), which is caused by the novel RNA virus called the severe acute respiratory syndrome coronavirus-2 (SARS-CoV-2), was first reported in December 2019. According to the World Health Organization COVID-19 dashboard, as of April 11, 2022, 490 million confirmed cases of COVID-19 and 6.1 million deaths due to COVID-19 have been reported worldwide [1]. Concurrently, 15 million confirmed cases of COVID-19 and 19,000 deaths due to COVID-19 are reported in the Republic of Korea [2]. Acute kidney injury (AKI) is the most common and major complication among the various complications associated with COVID-19 infection, and it has been reported to occur in approximately 10–50% of patients with COVID-19 [3,4,5,6]. AKI due to COVID-19 infection increases the risk of death. Thus, early detection and appropriate treatment of AKI are required to improve clinical outcomes [4,7].

Procalcitonin (PCT), which is a 116-amino acid precursor of calcitonin, has been identified as a useful inflammatory marker, not only in patients with sepsis but also in those with trauma or severe burns [8,9,10]. Presepsin (PSS), which is a 13-kDa cell surface polypeptide, is generated from the soluble cluster of differentiation 14 (CD14) and may be a useful marker for diagnosis, severity assessment, and predicting mortality in patients with sepsis [11,12].

Several studies have investigated the value of PCT and PSS in predicting AKI or hospital mortality in patients with COVID-19 [13,14,15,16]. However, to our knowledge, no studies have evaluated PCT and PSS together in predicting AKI and hospital mortality in patients with COVID-19 until now. The present study aimed to evaluate whether serum PCT and plasma PSS levels are associated with the prediction of AKI and 30-day hospital mortality in patients with COVID-19.

## 2. Materials and Methods

### 2.1. Study Design and Population

This retrospective cohort study of patients with COVID-19 presenting to the emergency department (ED) was performed at the Konkuk University Medical Center in Seoul, South Korea. It was conducted over a 25-month period, from February 2020 to February 2022.

The inclusion criteria were age >18 years and patients with COVID-19 who were admitted to our institution via the ED. The diagnosis of patients with COVID-19 was defined as a positive result using reverse transcription-polymerase chain reaction testing of a nasopharyngeal swab specimen. The exclusion criteria were patients who were transferred from other hospitals, those previously diagnosed with chronic kidney disease, and those who had insufficient medical or laboratory records. For patients who were hospitalized multiple times during the study period, the first hospitalization was included in the study.

The present study was performed in accordance with the Declaration of Helsinki and ethical requirements. The overall protocol of this study was approved by the Institutional Review Board of the hospital. The requirement for individual informed consent was waived by the review board.

### 2.2. Data Collection

All included patient demographics and clinical and laboratory data were retrieved from electronic medical records. The laboratory data set included white blood cell (WBC) count, blood urea nitrogen (BUN), creatinine, albumin, high-sensitivity C-reactive protein (hsCRP), lactate, PCT, and PSS levels. The confusion, urea, respiratory rate, blood pressure, and age ≥65 years (CURB-65) score and pneumonia severity index were assessed using the demographic, clinical, and laboratory data at ED admission.

The diagnosis of AKI was based on the Kidney Disease: Improving Global Outcomes clinical practice guidelines: an increase in serum creatinine of ≥0.3 mg/dL within 48 h or ≥50% from baseline within 7 days or urine output of <0.5 mL/kg/hour for six hours [17]. When reliable urine output data were not available, we only used the serum creatinine concentration. All patients who developed AKI were stratified into three categories according to the AKI stage (AKI stage 1, AKI stage 2, and AKI stage 3) using the highest serum creatinine concentration, urine output, and initiation of renal replacement therapy.

Venous blood samples were obtained from a peripheral vein at the same time in the ED. The blood samples were centrifuged and divided into small aliquots to avoid repeated freezing and thawing. Then, within 2 h from the collection, the samples were stored at −70 °C until use. Frozen samples were thawed at room temperature and gently mixed just before the measurement of biomarkers.

Serum PCT levels were measured using Atellica IM 1600 (Siemens Healthineers, Erlangen, Germany), with a measurable range from 0.02 ng/mL to 50 ng/mL. Plasma PSS levels were determined using the PATHFAST analyzer (LSI Medience Corporation, Tokyo, Japan). This chemiluminescent enzyme immunoassay has a measurable range of 0 pg/mL to 10,000 pg/mL.

The primary endpoints of the present study were the development of AKI and 30-day hospital mortality.

### 2.3. Statistical Analysis

The collected data were recorded using Microsoft 365 Excel (Microsoft, Redmond, WA, USA). All records of the included patients were anonymized prior to analysis. Statistical analyses were performed using IBM SPSS Statistics 27 (IBM Corp., Armonk, NY, USA) and MedCalc Version 20.027 (MedCalc Software, Ostend, Belgium).

The Kolmogorov–Smirnov test was used to determine whether the variables were normally distributed. Categorical variables were expressed as numbers and percentages, and the chi-square test or Fisher’s exact test was used for comparisons. Nonnormally distributed continuous variables were expressed as median (25th–75th percentile). The Mann–Whitney U test was used to compare two groups, whereas the Kruskal–Wallis test was used for multiple-group comparison. The prognostic values of biomarkers for predicting AKI and mortality were evaluated by constructing the receiver operating characteristic curves, and the area under the curve (AUC) and 95% confidence intervals (CIs) were obtained. The Youden index was used to establish the cutoff value. A logistic analysis was performed to evaluate the independent predictors of AKI development and 30-day hospital mortality; the odds ratios (ORs) and 95% CIs were calculated. Age, sex, comorbidities (diabetes mellitus, hypertension, chronic liver disease, chronic lung disease, cerebrovascular disease, and coronary artery disease), AKI, BUN, albumin, lactate, PCT, and PSS were included in the logistic regression analysis as potential confounding factors. Spearman correlation tests were performed to evaluate the relationships between WBC, albumin, hsCRP, lactate, PCT, and PSS. All statistical tests were two-sided, and *p <* 0.05 was considered statistically significant.

## 3. Results

### 3.1. AKI Predictive Value of PCT and PSS at ED Admission

A total of 151 patients who were diagnosed with COVID-19 during the study period were included in the present study. The median patient age was 77 (67–84) years, and 86 patients (57.0%) were male. AKI developed in 56 patients (37.1%), with AKI stage 1 in 31 patients (55.4%), stage 2 in 11 patients (19.6%), and stage 3 in 14 patients (25.0%). The clinical characteristics and outcomes of patients who developed AKI (patients with AKI) and those who did not (patients without AKI) are presented in Table 1. No differences in age, sex, comorbidities, heart rate, and body temperature were observed. Patients with AKI had higher CURB-65 scores and pneumonia severity indexes than those without AKI.

The serum PCT and plasma PSS levels at ED admission in patients with AKI were significantly higher than in those without AKI (*p <* 0.001; Table 2). The median levels of BUN, albumin, and lactate were also significantly higher in patients with AKI than in those without AKI (all *p <* 0.05). However, the WBC counts and hsCRP levels were not significantly different between patients with AKI and those without AKI.

The WBC count, BUN, albumin, hsCRP, lactate, PCT, and PSS at each AKI stage are presented in Table 3. The WBC count, albumin, hsCRP, lactate, PCT, and PSS levels did not differ significantly between the three AKI stages. The PSS level was 656 (347–1058) pg/mL in the AKI stage 1 group, 1302 (691–1639) pg/mL in the AKI stage 2 group, and 1695 (530–2847) pg/mL in the AKI stage 3 group. The median BUN level was significantly higher in the AKI stage 3 group than in the other two AKI groups.

The AUC value of serum PCT for predicting AKI in patients with COVID-19 was 0.811 (*p* < 0.001). The AUC values of BUN, albumin, lactate, PCT, and PSS for predicting AKI are presented in Table 4. Moreover, the AUC value of PCT for predicting AKI was significantly higher than those of BUN, albumin, lactate, and PSS (*p* = 0.022, *p* = 0.003, *p* = 0.005, and *p* = 0.045, respectively; Figure 1). The optimal cutoff values of PCT and PSS for predicting AKI in patients with COVID-19 were 2.26 ng/mL (sensitivity, 64.3%; specificity, 89.5%) and 572 pg/mL (sensitivity, 66.0%; specificity, 69.1%), respectively.

In multivariate analysis, BUN > 19.8 mg/dL (OR 3.695, 95% CI 1.431–9.540; *p* = 0.007), lactate > 1.79 mmol/L (OR 3.868, 95% CI 1.228–12.183; *p* = 0.021), PCT > 2.66 ng/mL (OR 5.455, 95% CI 1.787–16.649; *p* = 0.003), and PSS > 572 pg/mL (OR 2.508, 95% CI 1.041–6.040; *p* = 0.040) were identified as independent predictors of AKI development.

### 3.2. 30-Day Hospital Mortality Predictive Value of PCT and PSS at ED Admission

The 30-day hospital mortality rate was 12.6% (19/151) for all patients with COVID-19. The 30-day hospital mortality rate of patients with AKI was 21.4% (12/56), but that of patients without AKI was 7.4% (7/95; *p* = 0.020). Death occurred after a median of 6 (4–19) days of hospitalization. No differences were observed in age, sex, comorbidities, systolic blood pressure, heart rate, and respiratory rate between survivors and non-survivors (Table 5). Non-survivors had lower diastolic blood pressure and body temperature than survivors.

The median levels of PCT, PSS, BUN, and lactate in non-survivors were significantly higher than in survivors. The median albumin level in non-survivors was lower than in survivors. The WBC count and hsCRP were not significantly different between the survivors and non-survivors (Table 6).

The AUC values of BUN, albumin, lactate, PCT, and PSS for predicting 30-day hospital mortality are shown in Table 7. The AUC value of plasma PSS was 0.846 (95% CI: 0.774–0.903; *p <* 0.001). The AUC value of PSS was higher than that of PCT, but there was no statistically significant difference (*p* = 0.965; Figure 2). The AUC value of PSS was higher than those of BUN, albumin, and lactate (*p* = 0.011, *p* = 0.027, and *p* = 0.044, respectively). The optimal cutoff values of PSS and PCT for 30-day hospital mortality were 865 pg/mL (sensitivity, 84.6%; specificity, 76.0%) and 2.67 ng/mL (sensitivity, 68.4%; specificity, 77.3%), respectively.

In the univariate analysis, the following variables were significantly associated with 30-day hospital mortality: chronic liver disease (OR 4.762, 95% CI 1.039–21.838; *p* = 0.045), AKI (OR 3.429, 95% CI 1.261–9.319; *p* = 0.016), BUN > 17.7 mg/dL (OR 12.076, 95% CI 1.565–93.199, *p* = 0.017), albumin < 3.3 g/dL (OR 0.276, 95% CI 0.103–0.741; *p* = 0.011), lactate > 2.24 mmol/L (OR 4.989, 95% CI 1.382–18.017, *p* = 0.014), PCT > 2.67 ng/mL (OR 9.953, 95% CI 1.288–76.924, *p* = 0.028), and PSS > 865 pg/mL (OR 19.148, 95% CI 3.999–91.693; *p* < 0.001). In multivariate analysis, PSS > 865 pg/mL (OR 11.478, 95% CI 2.349–56.090; *p* = 0.003) was identified as an independent predictor of 30-day hospital mortality.

### 3.3. Correlation between PCT, PSS, and Other Laboratory Biomarkers

PCT levels were significantly positively correlated with PSS levels (Spearman’s rho = 0.463; *p <* 0.001; Figure 3). The Spearman correlation coefficients for the correlation of PCT levels with hsCRP and lactate levels were found to be 0.293, and 0.242, respectively (all *p <* 0.05). The correlation coefficients for the correlation of PSS levels with hsCRP and lactate levels were found to be 0.372, and 0.454, respectively (all *p <* 0.05). PCT and PSS levels were negatively correlated with plasma albumin levels (Spearman’s rho = −0.235; *p* = 0.004; Spearman’s rho = −0.404; *p <* 0.001, respectively). However, PCT and PSS levels showed no correlation with the WBC count (Spearman’s rho = 0.120; *p* = 0.144; Spearman’s rho = 0.010, *p* = 0.907, respectively).

## 4. Discussion

The present study evaluated the ability of serum PCT and plasma PSS levels at ED admission to predict AKI and 30-day hospital mortality. PCT and PSS levels at ED admission were significantly elevated in patients who developed AKI and those who died within 30 days of hospital admission.

PCT is a commonly used diagnostic and prognostic inflammatory marker in patients with sepsis in the ED. Recently, it has also been used to assess the severity and predict the prognosis in patients with COVID-19. A meta-analysis based on a total of 10 studies comprising 7716 patients with COVID-19 demonstrated that elevated PCT levels on hospital admission were strongly associated with high disease severity and hospital mortality [18]. The PCT levels were not affected by steroid administration and were significantly higher in patients with more severe COVID-19 and non-survivors [19]. Hu et al. [20] performed serial measurements of PCT levels in patients with COVID-19 and showed that a persistent increase in PCT levels was associated with a poor outcome. Unfortunately, the present study did not evaluate the relationship between inflammatory markers and COVID-19 disease severity. However, in accordance with previous findings, our study also showed that the serum PCT level was significantly higher in non-survivors than in survivors.

SARS-CoV-2 infection can trigger a cytokine-mediated hyperinflammatory response via the release of tumor necrosis factor-α or interleukin-6, which can, in turn, lead to an increase in PCT levels. Therefore, the serum PCT levels reflect the degree of cytokine secretion. The excessive cytokine release can lead to acute respiratory distress syndrome or extrapulmonary multiple organ failure and death [21]. Sayah et al. [19] reported that the AUC value of PCT for predicting mortality was 0.905 and the best cutoff value was 0.16 ng/mL, whereas the AUC value in the present study was 0.769 and the best cutoff value was 2.67 ng/mL. In another study, the AUC value of PCT for predicting COVID-19 mortality was 0.74 and the cutoff value was 0.1 ng/mL [15]. However, the best optimal cutoff value of PCT for predicting COVID-19 mortality has not been determined as yet.

The detailed mechanism underlying AKI occurrence in SARS-CoV-2 infection has not been fully elucidated. Angiotensin-converting enzyme 2 has been confirmed as the receptor for SARS-CoV-2 entry into human cells. Moreover, the expression of angiotensin-converting enzyme 2 in humans is nearly 100-fold higher in the kidney tubules than in the lung [22]. Thus, the kidney is possibly one of the main target organs of SARS-CoV-2 infection. In addition, direct invasion of the virus, cytokine storm, secondary infection with other viruses or bacteria, hypoperfusion, and nephrotoxic drugs may contribute to AKI development in patients with COVID-19.

The AKI incidence rate was 37.1% in the present study. Lim et al. [23] reported an AKI incidence rate of 18.3% in Korean patients. Some previous studies have reported that the incidence of AKI in patients with COVID-19 ranges from 7.2% to 11.5% [3,13,24]. However, other studies showed a higher AKI incidence rate of 44%–52.4% [4,5,6]. These various findings regarding AKI prevalence in patients with COVID-19 may be a result of the differences in various factors, such as disease severity, age of included patients, comorbidities, and hospital admission criteria.

Recent studies reported that elevated PCT levels (>0.1 ng/mL) were associated with AKI, and they gradually increased with an increase in the severity of AKI [3,14]. Wang et al. [13] reported that the PCT level was higher in the AKI group and the best cutoff value of PCT for predicting AKI was 0.1045 ng/mL, with a sensitivity of 92.9% and specificity of 72.3%. Similarly, in the present study, the median PCT level was significantly higher in patients who developed AKI than in those who did not. The AUC value for the PCT level ranged from 0.739 to 0.870. For a cutoff value of 2.26 ng/mL for PCT, the sensitivity and specificity for predicting AKI were 64.3% and 89.5%, respectively. The sensitivity in the present study was much lower than that reported by Wang et al., but the specificity was higher.

PSS is released into the circulation after proinflammatory response activation via the recognition of infection by the immune system [25]. It is a relatively novel inflammatory biomarker but a highly specific biomarker for the diagnosis of sepsis compared with interleukin-6 and PCT [26]. Several recent studies have shown that PSS is a useful biomarker for assessing the severity and predicting mortality in patients with COVID-19. In a study by Kocyigit et al. [27], PSS levels were significantly increased in the severe group compared with the mild and moderate groups of patients with COVID-19. In accordance with our findings, previous studies found that the PSS levels were significantly elevated in non-survivors than in survivors [16,28]. Park et al. [16] demonstrated that the AUC value of PSS for predicting 30-day mortality was 0.90 and that it was not significantly different from that of PCT. Similarly, in the present study, PSS (AUC value = 0.846) had an excellent prediction performance, which was higher than that of PCT (AUC value = 0.769), but there was no statistically significant difference (*p* = 0.965). Zaninotto et al. [29] and Keskinidou et al. [30] reported slightly lower performance of PSS than that observed in our study for predicting mortality in patients with COVID-19 admitted to the intensive care unit (AUC value = 0.72 and 0.83, respectively).

Interestingly, there was no difference in age between the survivors and non-survivors, and age was not an independent predictor of 30-day hospital mortality. These results are explained by the demographic characteristic that the age of the patients with COVID-19 in our study was 77 (67–84) years and 76.8% (116/151) were 65 years or older, which is older than the patients in other previous studies [6,15,19,24]. In addition, change in the severity of COVID-19 in admitted patients may be another reason. In the early period of the COVID-19 pandemic, patients with mild to moderate COVID-19 were hospitalized in our institution, but as the number patients with of COVID-19 surged, only elderly patients with very severe COVID-19 were hospitalized in our institution.

Few studies have investigated the value of PSS for predicting AKI in patients with COVID-19 and those without COVID-19. A recent study reported that the cutoff value of PSS for the prediction of sepsis-induced AKI was 708 pg/mL, with a sensitivity of 81.6% and specificity of 58.5%, in patients with sepsis who did not have COVID-19 and were admitted to the intensive care unit [31]. Mabrey et al. [32] found that patients with COVID-19 had significantly lower PSS levels than those presenting with other non-COVID-19 causes. In patients with COVID-19 only, elevated PSS levels were associated with a high risk of severe AKI. To our knowledge, the cutoff value of PSS for predicting AKI in patients with COVID-19 has not been studied until now. We found that the cutoff value of PSS for AKI in patients with COVID-19 was 572 pg/mL, and the sensitivity and specificity were 66.0% and 69.1%, respectively. The receiver operating characteristic curve analysis in the present study showed PCT to be a better predictor of AKI than PSS in patients with COVID-19 (AUC value = 0.811 vs. AUC value = 0.700; *p* = 0.045). The sensitivity of PSS for predicting AKI was slightly higher than that of PCT. However, the specificity of PCT for predicting AKI was significantly higher than that of PSS.

In addition, several studies found that serum albumin levels were lower in patients with AKI than in those without AKI [13,23,33]. In accordance with previous findings, non-survivors and patients who developed AKI showed low serum albumin levels in the present study. Moreover, we observed no differences in age and sex between patients with AKI and those without AKI. However, several previous studies reported more male and older patients among those with AKI [3,34].

The present study has limitations. First, it is a single-center retrospective study with a small sample size. Therefore, there is a possibility of selection bias in the inclusion of patients with COVID-19. Second, other biomarkers predicting AKI, such as cystatin C, kidney injury molecule-1, and neutrophil gelatinase-associated lipocalin, were not included in the study. Thus, they could not be compared with PCT and PSS. Third, PCT and PSS levels have limited sensitivity and specificity in predicting AKI and 30-day hospital mortality in patients with COVID-19. Nonetheless, to our knowledge, this is the first study to compare PCT and PSS in the prediction of AKI and hospital mortality in patients with COVID-19.

## 5. Conclusions

In conclusion, we confirmed that PCT and PSS levels were elevated in patients with AKI and non-survivors in patients with COVID-19. Initial PCT (>2.66 ng/mL) and PSS (>572 pg/mL) on ED admission were independent predictors of AKI development, and the level of PSS >865 pg/mL was identified as an independent predictor of 30-day hospital mortality. In the future, large prospective studies are needed to confirm our findings.

## Figures and Tables

**Figure 1 medicina-58-00727-f001:**
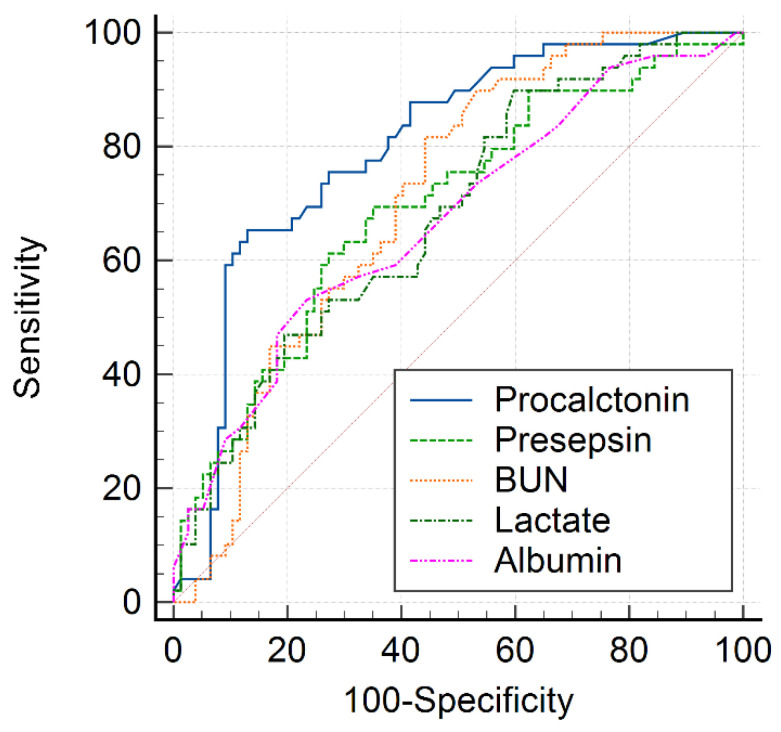
Receiver operating characteristic curves of laboratory biomarkers for acute kidney injury in patients with COVID-19.

**Figure 2 medicina-58-00727-f002:**
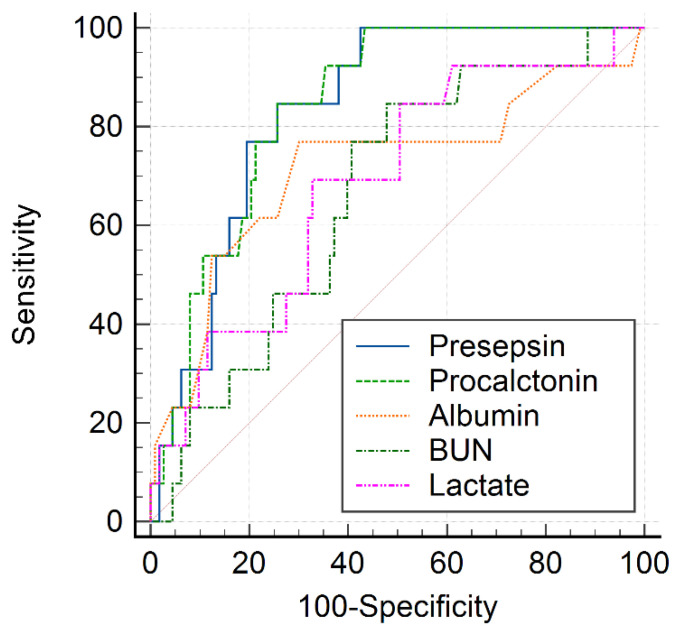
Receiver operating characteristic curves of laboratory biomarkers for 30-day hospital mortality in patients with COVID-19.

**Figure 3 medicina-58-00727-f003:**
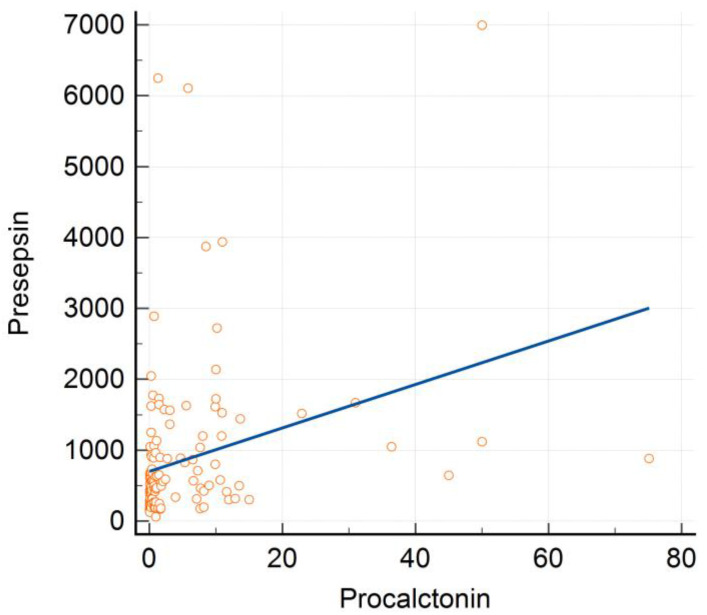
Correlation between procalcitonin and presepsin levels.

**Table 1 medicina-58-00727-t001:** Demographic and clinical characteristics of the included patients with COVID-19.

Variables	Overall (*n* = 151)	Patients without AKI (*n* = 95)	Patients with AKI (*n* = 56)	*p*-Value
Age, years	77 (67–84)	77 (67–84)	79 (65–85)	0.881
Male	86 (57.0)	58 (61.1)	28 (50.0)	0.234
Comorbidity				
Diabetes mellitus	42 (27.8)	27 (28.4)	15 (26.8)	0.853
Hypertension	73 (48.3)	45 (47.4)	28 (50.0)	0.866
Chronic liver disease	8 (5.3)	3 (3.2)	5 (8.9)	0.147
Chronic lung disease	10 (6.6)	4 (4.2)	5 (10.7)	0.174
Cerebrovascular disease	49 (32.5)	30 (31.6)	19 (33.9)	0.857
Coronary artery disease	7 (4.6)	6 (6.3)	1 (1.8)	0.260
Vital sign				
Systolic blood pressure, mmHg	115 (97–132)	116 (100–139)	110 (88–127)	0.012
Diastolic blood pressure, mmHg	66 (56–77)	67 (57–82)	61 (53–71)	0.001
Heart rate, beats/min	100 (86–116)	100 (86–114)	100 (87–118)	0.318
Respiratory rate, breaths/min	20 (20–23)	20 (19–23)	21 (20–25)	0.028
Body temperature, °C	37.2 (36.6–38.2)	37.4 (36.6–38.1)	37.2 (36.7–38.3)	0.774
CURB-65 score	2 (1–3)	2 (1–2)	2 (1–3)	0.045
Pneumonia severity index	100 (81–126)	94 (78–120)	110 (87–141)	0.006
Death	19 (12.6)	7 (7.4)	12 (21.4)	0.020

Values are presented as median (25–75% interquartile range) or number (%). COVID-19: coronavirus disease 2019, AKI: acute kidney injury, CURB-65: confusion, urea, respiratory rate, blood pressure, age ≥ 65 years.

**Table 2 medicina-58-00727-t002:** Comparison of laboratory results on ED admission between patients with COVID-19 with AKI and those without AKI.

Variable	Patients without AKI (*n* = 95)	Patients with AKI (*n* = 56)	*p*-Value
WBC count, ×10^3^/μL	10,920 (7365–15,040)	11,150 (7465–14,945)	0.331
BUN, mg/dL	18.6 (12.1–33.0)	31.8 (21.0–51.0)	<0.001
Creatinine, mg/dL	0.72 (0.56–0.84)	0.73 (0.61–0.92)	0.392
Albumin, g/dL	3.6 (3.4–3.9)	3.3 (2.9–3.7)	0.004
hsCRP, mg/dL	9.50 (4.65–20.55)	13.01 (6.49–23.6)	0.117
Lactate, mmol/L	2.12 (1.56–3.38)	3.30 (2.00–5.74)	0.003
Procalcitonin, ng/mL	0.29 (0.06–1.35)	6.62 (0.89–9.99)	<0.001
Presepsin, pg/mL	447 (239–658)	801 (442–1589)	<0.001

Values are presented as median (25–75% interquartile range). ED: emergency department, COVID-19: coronavirus disease 2019, AKI: acute kidney injury, WBC: white blood cell, BUN: blood urea nitrogen, hsCRP: high-sensitivity C-reactive protein.

**Table 3 medicina-58-00727-t003:** Comparison of laboratory results according to AKI severity.

Variable	AKI Stage 1 (*n* = 31)	AKI Stage 2 (*n* = 11)	AKI Stage 3 (*n* = 14)	*p*-Value
WBC count, ×10^3^/μL	11,150 (7090–15,530)	9750 (6048–20,593)	11,400 (8235–14,816)	0.721
BUN, mg/dL	25.2 (18.6–42.1)	26.3 (23.0–33.2) ^c^	56.0 (45.0–71.9) ^b^	0.002
Albumin, g/dL	3.4 (3.1–3.7)	3.5 (2.9–3.7)	3.1 (2.7–3.5)	0.554
hsCRP, mg/dL	11.23 (3.44–21.52)	14.2 (6.4–23.6)	21.8 (11.4–29.9)	0.151
Lactate, mmol/L	2.45 (1.97–5.37)	3.70 (2.09–5.90)	3.67 (2.48–6.26)	0.588
Procalcitonin, ng/mL	7.62 (0.93–10.75)	4.28 (0.56–8.28)	4.64 (0.79–9.96)	0.581
Presepsin, pg/mL	656 (347–1058) ^b^	1302 (691–1639) ^a^	1695 (530–2847)	0.078

Values are presented as median (25–75% interquartile range). AKI: acute kidney injury, WBC: white blood cell, BUN: blood urea nitrogen, hsCRP: high-sensitivity C-reactive protein. ^a^
*p* < 0.05, vs. stage 1; ^b^
*p* < 0.05, vs. stage 2; ^c^
*p* < 0.05, vs. stage 3.

**Table 4 medicina-58-00727-t004:** The value of laboratory biomarkers for predicting AKI in patients with COVID-19.

Variable	AUC	95% CI	Cutoff Value	Sensitivity (%)	Specificity (%)	*p*-Value
BUN	0.659	0.582–0.724	>19.8	83.9	62.1	<0.001
Albumin	0.641	0.559–0.717	<3.2	42.9	82.1	0.003
Lactate	0.649	0.564–0.728	>1.79	87.0	37.7	0.002
Procalcitonin	0.811	0.739–0.870	>2.26	64.3	89.5	<0.001
Presepsin	0.700	0.615–0.776	>572	66.0	69.1	<0.001

AKI: acute kidney injury, COVID-19: coronavirus disease 2019, AUC: area under the curve, CI: confidence interval, BUN: blood urea nitrogen.

**Table 5 medicina-58-00727-t005:** Comparison of demographic and clinical characteristics between survivors and non-survivors among patients with COVID-19.

Variable	Survivors (*n* = 132)	Non-Survivors (*n* = 19)	*p*-Value
Age, years	77 (67–85)	77 (66–83)	0.682
Male	74 (56.1)	12 (63.2)	0.627
Comorbidity			
Diabetes mellitus	38 (28.8)	4 (21.1)	0.592
Hypertension	64 (48.5)	9 (47.4)	0.927
Chronic liver disease	5 (3.8)	3 (15.8)	0.063
Chronic lung disease	9 (6.8)	1 (5.3)	0.799
Cerebrovascular disease	45 (34.1)	4 (21.1)	0.305
Coronary artery disease	7 (5.3)	0 (0)	0.597
Vital sign			
Systolic blood pressure, mmHg	116 (99–133)	93 (88–119)	0.243
Diastolic blood pressure, mmHg	66 (56–79)	53 (51–65)	0.023
Heart rate, beats/min	100 (89–116)	87 (83–117)	0.726
Respiratory rate, breaths/min	20 (20–23)	20 (19–23)	0.076
Body temperature, °C	37.4 (36.7–38.4)	36.5 (36.0–37.0)	0.002
CURB-65 score	2 (1–3)	3 (2–4)	<0.001
Pneumonia severity index	97 (80–120)	138 (125–178)	<0.001
Acute kidney injury	44 (33.3)	12 (63.2)	0.020

Values are presented as median (25–75% interquartile range) or number (%). COVID-19: coronavirus disease 2019, CURB-65: confusion, urea, respiratory rate, blood pressure, age ≥ 65.

**Table 6 medicina-58-00727-t006:** Comparison of laboratory results on ED admission between survivors and non-survivors among patients with COVID-19.

Variable	Survivors (*n* = 132)	Non-Survivors (*n* = 19)	*p*-Value
WBC count, × 10^3^/μL	10,700 (7390–14,930)	12,100 (6705–17,890)	0.365
BUN, mg/dL	21.8 (15.7–39.7)	29.8 (24.4–57.8)	0.016
Albumin, g/dL	3.6 (3.2–3.9)	2.9 (2.7–3.6)	0.004
hsCRP, mg/dL	11.65 (4.80–21.81)	19.07 (5.03–30.88)	0.194
Lactate, mmol/L	2.32 (1.72–3.99)	3.33 (2.31–6.82)	0.034
Procalcitonin, ng/mL	0.70 (0.09–3.51)	9.98 (4.42–12.32)	< 0.001
Presepsin, pg/mL	495 (258–845)	1441 (960–1935)	< 0.001

Values are presented as median (25–75% interquartile range). ED: emergency department, COVID-19: coronavirus disease 2019, WBC: white blood cell, BUN: blood urea nitrogen, hsCRP: high-sensitivity C-reactive protein.

**Table 7 medicina-58-00727-t007:** The value of laboratory biomarkers for predicting 30-day hospital mortality in patients with COVID-19.

Variable	AUC	95% CI	Cutoff Value	Sensitivity (%)	Specificity (%)	*p*-Value
BUN	0.671	0.590–0.746	>17.7	94.7	40.2	0.004
Albumin	0.702	0.622–0.773	<3.3	68.4	71.2	0.009
Lactate	0.652	0.567–0.731	>2.24	84.2	48.3	0.021
Procalcitonin	0.769	0.694–0.834	>2.67	68.4	77.3	<0.001
Presepsin	0.846	0.774–0.903	>865	84.6	76.0	<0.001

COVID-19: coronavirus disease 2019, AUC: area under the curve, CI: confidence interval, BUN: blood urea nitrogen.

## Data Availability

The data that support the findings of this study are available on request to the corresponding author.

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
