# Peer review of "Predictive Values of Procalcitonin and Presepsin for Acute Kidney Injury and 30-Day Hospital Mortality in Patients with COVID-19"

_medicina, 2022, doi:10.3390/medicina58060727_

Round 1
Reviewer 1 Report
Thank you for giving me the opportunity to review the manuscript titled
Thank you for submitting a manuscript of "Predictive Values of Procalcitonin and Presepsin for Acute Kidney Injury and 30-Day Hospital Mortality in Patients with COVID-19"
This is a very interesting topic. but there are several concerns.
- The authors suggested that Procalcitonin and Presepsin correlated with COVID-19-related mortality and morbidity. However, those correlations were already studied in previous reports (DOI: 10.1016/j.ijid.2022.02.054;,DOI: 10.3390/medicina57101070; DOI: 10.1002/jcla.23805). Would you address any strengths over the previous studies?
- Would you describe the time when AKI was diagnosed?
- Would you suggest Laboratory values representing renal function such as GFR or Creatinine?
- Would you compare the levels of Procalcitonin and Presepsin in patients with normal kidney function, AKI stage1, AKI stage 2, and AKI stage 3?
- The authors suggested AUCs and P-values of laboratory biomarkers for predicting 30-day hospital mortality in patients with COVID-19. Could you conduct a multivariate analysis of risk factors associated with AKI or mortality? The analysis shows whether Procalcitonin or Presepsin is an independent factor.
Author Response
1. The authors suggested that Procalcitonin and Presepsin correlated with COVID-19-related mortality and morbidity. However, those correlations were already studied in previous reports (DOI:10.1016/j.ijid.2022.02.054;,DOI:10.3390/medicina57101070; DOI: 10.1002/jcla.23805). Would you address any strengths over the previous studies?
Reply: Procalcitonin has been extensively studied as an inflammatory biomarker and is now widely used in clinical practice. By contrast, presepsin is a relatively novel biomarker. Several studies have shown that presepsin not only is useful for the diagnosis of sepsis but could also be a predictor of the severity and mortality of the disease. Previous reports have studied only one marker, either procalcitonin or presepsin, for predicting AKI or mortality in patients with COVID-19. By contrast, we studied two biomarkers simultaneously in predicting AKI and mortality in patients with COVID-19. These points are the strength of our report.
2. Would you describe the time when AKI was diagnosed?
Reply: The diagnosis of AKI in our study was based on the KDIGO practice guideline; increase in serum creatinine of ≥ 0.3 mg/dL within 48 hours or ≥ 50% from baseline within 7 days or urine output of < 0.5 mL/kg/hour for six hours. We added this information in Section 2.2.
3. Would you suggest laboratory values representing renal function such as GFR or creatinine?
Reply: There were no differences in initial creatinine levels on ED admission between the non-AKI and AKI groups (0.72 [0.56–0.84] vs 0.73 [0.61–0.92], respectively; p = 0.392). We added this information in Section 3 (Table 2).
4. Would you compare the levels of procalcitonin and presepsin in patients with normal kidney function, AKI stage1, AKI stage 2, and AKI stage 3?
Reply: Procalcitonin and presepsin levels were different between the patients with non-AKI and those with AKI. Because this difference can be seen as an increase in procalcitonin and presepsin levels according to the AKI stage, we compared procalcitonin and presepsin levels according to the AKI stage only in the AKI group, excluding the non-AKI group. The procalcitonin and presepsin levels did not differ significantly between the three AKI stages. We added this information in Section 3 (Table 3).
5. The authors suggested AUCs and P-values of laboratory biomarkers for predicting 30-day hospital mortality in patients with COVID-19. Could you conduct a multivariate analysis of risk factors associated with AKI or mortality? The analysis shows whether procalcitonin or presepsin is an independent factor.
Reply: In multivariate analysis, BUN, lactate, procalcitonin, and presepsin were independent predictors of AKI development. In addition, presepsin was the only independent predictor of 30-day hospital mortality. We have added these results in Section 3. There are many tables in the manuscript, so it is presented descriptively.
Reviewer 2 Report
I would like to congratulate the authors for the work entitled “Predictive values of procalcitonin and presepsin for acute kidney Injury and 30-day Hospital Mortality in patients with COVID-19”. This is a very interesting work about COVID-19 infection and the value of these biomarkers in prognostic of these patients. Next, only two comments on this work that I think that help to improve the quality of this manuscript.
In section 3.3 of results, a figure with the results of the spearman correlation should be added to clarify these results.
Surprisingly, regarding mortality there were no differences in the age of the patients. I do not know how the authors interpret this result and how they justify it. This aspect should be added to the discussion.
Author Response
1. In section 3.3 of results, a figure with the results of the spearman correlation should be added to clarify these results.
Reply: We added Figure 3 (Correlation between procalcitonin and presepsin levels) to clarify the results.
2. Surprisingly, regarding mortality there were no differences in the age of the patients. I do not know how the authors interpret this result and how they justify it. This aspect should be added to the discussion.
Reply: In our study, the majority of enrolled patients were relatively older than in previous studies. In addition, the severity of COVID-19 in hospitalized patients has changed over time during the COVID-19 pandemic. These factors may have influenced the results of our study. We added this information in Section 4.
Round 2
Reviewer 1 Report
Thank you for the invitation to review this study.
1. The authors studied two biomarkers simultaneously in predicting AKI and mortality in patients with COVID-19. However, it seems that there is not enough explanation for the advantages of simultaneous measurement. Also, comparing those biomarkers in predicting mortality was already studied in a previous report (https://doi.org/10.3343/alm.2022.42.4.406).
2. Conclusion: Please revise this section to concentrate on your data focusing on your results. And please address the "limited sensitivity and specificity" in the limitation part of the Discussion.
Author Response
1. The authors studied two biomarkers simultaneously in predicting AKI and mortality in patients with COVID-19. However, it seems that there is not enough explanation for the advantages of simultaneous measurement. Also, comparing those biomarkers in predicting mortality was already studied in a previous report (https://doi.org/10.3343/alm.2022.42.4.406).
Reply: Thank you very much for your kind comment. Although the usefulness of procalcitonin and presepsin inpatients with COVID-19 has been studied, the characteristics of the enrolled subjects in each study were different. In contrast, our study is meaningful as it compared procalcitonin and presepsin in same study subjects. Also, in the report you mentioned, the VACO index was added to the study, but it included a relatively small number of patients (n = 54) and only studied 30-day hospital mortality. On the other hand, our study was conducted on 151 COVID-19 patients to predict 30-day hospital mortality and acute kidney injury. We ask for your kind understanding.
2. Conclusion: Please revise this section to concentrate on your data focusing on your results. And please address the "limited sensitivity and specificity" in the limitation part of the Discussion.
Reply: Thank you for your comment. As you mentioned, we added the ‘limited sensitivity and specificity’ in the limitation part. Also, we modified conclusion to be: 1) PCT and PSS levels were elevated in patients with AKI and nonsurvivors, 2) PCT and PSS were independent predictors for predicting AKI, 3) PSS was independent predictor for predicting 30-day hospital mortality.
